# A graph-based algorithm for RNA-seq data normalization

**Diem-Trang Tran**[1]*, **Aditya Bhaskara**[1], **Balagurunathan Kuberan**[2,3], **Matthew Might**[4]

**1** School of Computing, University of Utah, Salt Lake City, Utah, United States of America, **2** Department of Medicinal Chemistry, University of Utah, Salt Lake City, Utah, United States of America, **3** Department of Biology, University of Utah, Salt Lake City, Utah, United States of America, **4** Hugh Kaul Precision Medicine Institute, University of Alabama at Birmingham, Birmingham, Alabama, United States of America

* dtrang.tran@utah.edu

**Data Availability Statement:** The data used in this study were originally generated under the ENCODE project and made publicly available on the project website, https://www.encodeproject.org/. This study used a compilation of human and mouse tissue mRNA-seq assays that were pre-processed

## Abstract

The use of RNA-sequencing has garnered much attention in recent years for characterizing and understanding various biological systems. However, it remains a major challenge to gain insights from a large number of RNA-seq experiments collectively, due to the normalization problem. Normalization has been challenging due to an inherent circularity, requiring that RNA-seq data be normalized before any pattern of differential (or non-differential) expression can be ascertained; meanwhile, the prior knowledge of non-differential transcripts is crucial to the normalization process. Some methods have successfully overcome this problem by the assumption that most transcripts are not differentially expressed. However, when RNA-seq profiles become more abundant and heterogeneous, this assumption fails to hold, leading to erroneous normalization. We present a normalization procedure that does not rely on this assumption, nor prior knowledge about the reference transcripts. This algorithm is based on a graph constructed from intrinsic correlations among RNA-seq transcripts and seeks to identify a set of densely connected vertices as references. Application of this algorithm on our synthesized validation data showed that it could recover the reference transcripts with high precision, thus resulting in high-quality normalization. On a realistic data set from the ENCODE project, this algorithm gave good results and could finish in a reasonable time. These preliminary results imply that we may be able to break the long persisting circularity problem in RNA-seq normalization.

## Introduction

RNA-sequencing (RNA-seq) has become a critical tool to study biological systems [1]. The technique starts with extracting the RNA fraction of interest and preparing them for high-throughput sequencing. Sequencers typically output short reads that are then assembled or aligned to a pre-assembled genome or transcriptome, resulting in a quantity called *read count* for each transcript. Due to variations in sequencing depth (i.e. library size, the total number of read count per sample) and in relative contribution of each transcript under different conditions, these read counts need to be normalized such that the changes in their measurements, usually indicated by fold-change, accurately reflect the differences between conditions. This is

into read counts. The data from 71 mouse tissue mRNA-seq assays, derivative data sets and the necessary information to generate them as described in this work are available at doi:10.5281/zenodo.2667313. The data from 26 human and mouse tissues were downloaded from http://doi.org/10.5281/zenodo.17606. The analyses are available as R notebooks, with source code and pre-compiled HTML file deposited at doi:10.5281/zenodo.3472166.

**Funding:** This material was based on research supported by the National Heart, Lung, and Blood Institute (NHLBI)–NIH sponsored Programs of Excellence in Glycosciences [grant number HL107152 to B.K.], and by the National Science Foundation [CAREER grant 1350344 to M.M.]. The U.S. Government is authorized to reproduce and distribute reprints for Governmental purposes notwithstanding any copyright notation thereon.

often named the between-sample normalization problem, which has attracted much efforts in solving it. One universal aim of the methods proposed in this arena is to derive sample-specific scaling factors, either by a data-driven or knowledge-based approach. Data-driven methods to estimate normalization factors have evolved from simple total count (TC) to more sophisticated statistics [2–4], to iterative approaches [5–8] that are built on top of simpler estimators. The most primitive estimator, usually referred to as total count (TC) normalization, simply uses the sum of all counts in each sample to normalize every gene. This operation is in fact a within-sample normalization. First-generation methods of between-sample normalization involve a closed-form evaluation of one scaling factor per sample. In the upper quartile (UQ) method [2], each samples are scaled by the total counts of genes in the upper quartile after removing zero counts. Adjustments based on more sophisticated statistics such as median of the ratio of a gene and its geometric mean across samples [4] or Trimmed Mean of M-values (TMM) [3] worked on the assumption that most genes are not differentially expressed across conditions. Second-generation methods usually follow an iterative scheme in which read counts are first normalized under the hypothesis that (almost) no genes are differentially expressed (DE), the set of non-DE genes are then refined based on some gene-based measures, and the procedure repeats until convergence. PoissonSeq [5] starts with total count as initial estimate of scaling factor, equivalently the result of fitting a Poisson log linear model under the null hypothesis (no gene is DE), then calculates the goodness-of-fit for each genes, marking those within a chosen quantile to be non-DE, and fits the model again under new hypothesis (selected genes are non-DE). TbT [6] and its generalized form DEGES (DE-Gene Elimination Strategy) [7] calculate normalization factors with a first-generation method (UQ, TMM, DESeq), tests for differential expression by a DE detection routine (edgeR, DESeq, baySeq), removes the DE genes and re-calculates normalization factors. In a similar fashion, a more recent method by Zhuo et al. [8] starts with the DESeq normalization (sometimes referred to as the relative log expression (RLE) method, to distinguish the scaling step from the DE detection step of DESeq), fitting a Poisson log linear model for each gene, calculating the total variance of each genes under this model, selecting a subset with least variance as the references, updating the normalization factors and repeats if necessary. Knowledge-based methods rely on sources of information beyond the RNA-seq measurements to determine possibly invariant transcripts and pivot on them to normalize the raw counts. Following the common practices in quantitative PCR, housekeeping genes, believed to be expressed at similar levels across conditions, have been used as endogenous references [9]. However, many common housekeeping references turned out to vary significantly (see Huggett et al. [10] for an extensive list of such examples), leading to the favor of exogenous controls, i.e spike-in RNAs [11, 12]. The addition of external spike RNAs significantly increases the cost, complicates experimental processes, and is inapplicable for integrating the large number of data from different experiments and laboratories which used different spikes, or most of the times, no spike at all. At the intersection between data-driven and knowledge-based approaches, a hybrid method proposed by Chen et al. [13] uses functional annotations to calculate the relevance of a genes with respect to the target experimental conditions and automatically suggest most functionally distant set of genes as references.

Among data-driven methods, early benchmark studies have shown that some first-generation methods can perform very well in differential expression (DE) analyses [14, 15]. A major caveat is the core assumption that most genes are not differentially expressed across conditions of interest [16]. This assumption fails to hold when one needs to analyze multitude and variable conditions. Second-generation, iterative procedures consciously operate under a circular dependence wherein the pruning of DE genes depends on a good normalization [8]. They are, in turn, dependent on the same assumption as that of the first-generation methods and tend to

exaggerate the errors when the proportion of DE genes are high [16]. Knowledge-based methods suffer from a different type of circularity, implicitly requiring that new findings align with previously documented gene functions. Although useful, they may become problematic when experimental conditions are vastly different from established ones.

We propose a new method of normalization that can break this circularity, without relying on assumption of biological similarity between the conditions, nor *a priori* knowledge about the controls, internal or external. We first show that there exist intrinsic correlations among reference transcripts that could be exploited to distinguish them from differential ones, and then introduce an algorithm to discover these references. This algorithm works by modeling each transcript as a vertex, and correlations between them as edges in a graph. Under this model, a set of references manifest themselves as a complete subgraph and therefore can be identified by solving a clique problem. With a few practical adjustments, the algorithm can be finished in reasonable time and give good results on both the validation data and a real data set.

## Formulation of graph-based normalization

### Definitions and notations

An RNA-seq measurement on one biological sample results in a vector of abundance values of $n$ genes/transcripts. A collection of measurements on $m$ samples results in $m$ such vectors can then be represented by an $m \times n$ matrix.

Let $A$ denote the abundance matrix in which the element $a_{ij}$ is the true abundance of transcript $j$ in sample $i$, $C$ the read count matrix in which the element $c_{ij}$ represents the read count of transcript $j$ in sample $i$. The total read counts of row $C_i$ is the sequencing depth (or library size) of sample $i$, $M_i = \sum_j c_{ij}$. Let $A_{rel}$ denote the relative abundance matrix, of which the element $a_{ij}^{rel}$ is the relative abundance of transcript $j$ in sample $i$

$$a_{ij}^{rel} = \frac{a_{ij}}{\sum_j a_{ij}} \quad , \quad \sum_j a_{ij}^{rel} = 1$$

$A$ is the underlying expression profile dictating the measurement in $C$. Since the exact recovery of $A$ is difficult, it usually suffices to normalize $C$ to a manifest abundance matrix $A^*$ of which $a_{ij}^* = \text{const} \times a_{ij}$, const is an unknown, yet absolute constant. Such $A^*$ is considered a *desirable* normalization.

**Differential** genes/transcripts are differentially expressed due to distinct biological regulation of the conditions being studied, thus are of biological interest. The term *condition* may represent different states of a cell population (normal vs tumor, control vs drug-administered, etc.), or different histological origins (cerebellum vs frontal cortex, lung, liver, muscle, etc.). Although *differential* is usually encountered in the comparison of two conditions, we will use the term with a broader sense, for multi-condition assays to indicate genes/transcripts that vary with the conditions.

**Reference** genes/transcripts are expressed at equivalent levels across conditions. From the biological standpoint, reference genes/transcripts should be constitutive, or at the least, are not under the biological regulations that distinguish the conditions. It should be noted that, for the purpose of normalizing read counts, reference genes/transcripts are numerically stable across conditions, and are not necessarily related to housekeeping genes/transcripts.

In this text, it is often more appropriate to use the term **feature** in place of **gene/transcript** to indicate the target entity of quantification, which can be mRNA, non-coding RNA or spike-in RNA. These features can be quantified at the transcript level or the gene level which aggregates the abundance of multiple isoforms if necessary.

## Normalization by references

Normalization by references has been a standard practice since the early expression profiling experiments where a few transcripts are measured individually by quantitative PCR [17]. This idea has been carried over to normalizing RNA-seq expression profiles. Here we show how it works in this new setting (Theorem 1), and how the inclusion of differential features or exclusion of reference ones affect the normalization (Proposition 1 and 2).

In the following, $\sum_{j \in ref}$ means summation over features in the reference set, and $\sum_{j \in all}$ means summation over all features.

For simplicity, feature abundance is treated as condition-specific constant, that is, assuming biological and technical variance across replicates of the same condition is zero. The operations remain the same when abundance levels and read counts are treated as random variables, by replacing the relevant constants ($a_{ij}$, $c_{ij}$) by the expectations of their random-variable counterparts ($\mathbb{E}[a_{ij}], \mathbb{E}[c_{ij}]$).

**Theorem 1.** *Let $N = [N_1, N_2, \ldots, N_m]^T$ be the reference-based normalizing vector, i.e. the scaling factor $N_i$ is the sum of read counts of all the reference features in that sample.*

$$N_i = \sum_{j \in ref} c_{ij}$$

*The manifest abundance $A^*$ resulted from normalizing C against N is the desirable manifest abundance. In other words, if $a_{ij}^* = c_{ij}/N_i$ then $a_{ij}^* = const \times a_{ij}$, const is a quantity that does not depend on the row/column.*

*Proof.*

$$a_{ij}^* = \frac{c_{ij}}{N_i} \tag{1}$$

$$= \frac{M_i \times a_{ij}^{rel}}{N_i} \quad \text{(share of read count is proportional to relative abundance)} \tag{2}$$

$$= \frac{M_i \times a_{ij}^{rel}}{\sum_{k \in ref} c_{ik}} \quad \text{(by definition of reference} - \text{based normalizing factors)} \tag{3}$$

$$= \frac{M_i}{M_i \sum_{k \in ref} a_{ik}^{rel}} \times a_{ij}^{rel} \quad \text{(by assumption about read distribution)} \tag{4}$$

$$= \frac{a_{ij}^{rel}}{\sum_{k \in ref} a_{ik}^{rel}} \tag{5}$$

$$= \frac{a_{ij}}{\sum_{k \in ref} a_{ik}} \tag{6}$$

Since references are constant across samples, their sum is also constant.

$$a_{1j} = a_{2j} = \cdots = a_j$$

$$\sum_{j \in ref} a_{ij} = \sum_{j \in ref} a_j = A^{ref}$$

Thus, the Eq (6) implies that the manifest abundance is simply a multiple of the true abundance.

Note that in the proof above, we have used a simplified assumption about the distribution of read counts, $c_{ij} = a_{ij}^{rel} \times M_i$. This assumption has abstracted away length bias in which a longer transcript takes a larger share of reads. To account for length more explicitly, one can define a quantity called length-adjusted relative abundance, $b_{ij}^{rel} = \frac{a_{ij}\ell_j}{\sum_{g\in all} a_{ig}\ell_g}$, in which $\ell_j$ is the length correction factor for feature $j$. The model of read counts is now equivalent to that used by Robinson & Oshlack [3], i.e.

$$c_{ij} = \frac{a_{ij}\ell_j M_i}{\sum_{g\in all} a_{ig}\ell_g} = b_{ij}^{rel} \times M_i$$

The relative abundance $a_{ij}^{rel}$ can then be replaced by $b_{ij}^{rel}$, resulting in a proof identical to the above.

**A subset of references is sufficient for normalization.** Proposition 1 and 2 demonstrate the effect of mistaking differential features in the reference set, or missing some reference features during normalization.

**Proposition 1**. *Inclusion of any differential feature in the normalizing factors leads to an invalid normalization.*

*Proof.* Let $a_{ij}' = \frac{c_{ij}}{N_i + c_{id}}$ in which $c_{id}$ is the count of a differential feature, in similar operations as the above, we can arrive at $a_{ij}' = \frac{a_{ij}}{\sum_j^{ref} a_{ij} + a_{id}}$. Since $d$ is differential gene, the denominator is not constant, thus $A' \neq const \times A$.

**Proposition 2**. *Any non-empty subset of the reference set leads to a valid normalization.*
*Proof.* Identical to that of Proposition 1.

Intuitively, a larger reference set is more favorable than a smaller one, due to the noisy nature of experimental measurements. In practice, a larger set of references can be identified with more confidence than a smaller one, as elaborated later in *Practical considerations*.

## Manifest correlation of transcripts in RNA-seq

In the following sections we use $c$ for read count and $a$ for the true abundance, hence $c$ is a function of $a$, i.e. $c = f(a)$. The subscript $i, j$ indicates different conditions, and $u, v$ different features.

**$u$ and $v$ are both references.** By definition, reference features are constant across conditions, i.e.

$$\begin{cases} a_{u,i} = a_{u,j} = a_u & \forall i,j \text{ such that } i \neq j \\ a_{v,i} = a_{v,j} = a_v & \forall i,j \text{ such that } i \neq j \end{cases} \tag{7}$$

Since read counts depends on true abundance $a$, sequencing depth $M$, and transcript length $\ell$,

$$\begin{cases} c_{u,i} \propto a_u \cdot M_i \cdot \ell_u \\ c_{v,i} \propto a_v \cdot M_i \cdot \ell_v \end{cases} \Rightarrow \frac{c_{u,i}}{c_{v,i}} = \frac{a_u \cdot \ell_u}{a_v \cdot \ell_v} \tag{8}$$

$$\text{Similarly with condition } j, \quad \frac{c_{u,j}}{c_{v,j}} = \frac{a_u \cdot \ell_u}{a_v \cdot \ell_v} \tag{9}$$

From Eqs (8) and (9), it is true that

$$\frac{c_{u,i}}{c_{v,i}} = \frac{c_{u,j}}{c_{v,j}} = \frac{c_u}{c_v} = \text{const}$$

**Observation 1**. *If u and v are both reference features, their read counts are linearly correlated.*

**u is differential, v is reference (or vice versa).** Equivalently,

$$\begin{cases} a_{u,i} = a_{u,j} = a_u & \forall i, j \text{ such that } i \neq j \\ a_{v,i} \neq a_{v,j} \end{cases}$$

With similar operation, the observed relation between $u$ and $v$ in each condition are

$$\frac{c_{u,i}}{c_{v,i}} = \frac{a_u \cdot \ell_u}{a_{v,i} \cdot \ell_v}$$

$$\frac{c_{u,j}}{c_{v,j}} = \frac{a_u \cdot \ell_u}{a_{v,j} \cdot \ell_v}$$

**Observation 2**. *If u is a differential feature and v is a reference one (or vice versa), their read counts are not linearly correlated.*

**u and v are both differential.** Equivalently,

$$\begin{cases} a_{u,i} \neq a_{u,j} \\ a_{v,i} \neq a_{v,j} \end{cases}$$

Linear correlation in this case requires that $\dfrac{c_{u,i}}{c_{v,i}} = \dfrac{c_{u,j}}{c_{v,j}} \Leftrightarrow \dfrac{a_{u,i}}{a_{v,i}} = \dfrac{a_{u,j}}{a_{v,j}}$

**Observation 3**. *If u and v are both differential features, the two will exhibit linear correlation if and only if they vary at similar proportion (i.e. same fold change) across different conditions.*

## Graph-based normalization algorithm

It follows from the earlier remarks (Observations 1, 2, 3) that all the references in an RNA-seq data set are linearly correlated with one another. Although the derivation was based on a simplistic treatment of expression levels as condition-specific constants, the effect was in fact observed in real data. In the ENCODE data set where ERCC spike-in RNAs were added at constant concentrations across samples, the read counts of these spike-ins are highly correlated (Fig 1).

On this premise, using a graph that models features as vertices, and positive correlation between them as edges, it is apparent that reference features will manifest themselves as a complete subgraph. However, there might exist other complete subgraphs composed of strongly co-expressed differential features (Observation 3). A couple of criteria can be employed to further distinguish the reference subgraph and the differential ones. First, differential subgraphs represent tightly regulated and co-expressed features in biological systems throughout all conditions. Consequently, larger differential subgraph is less likely to exists. By setting the

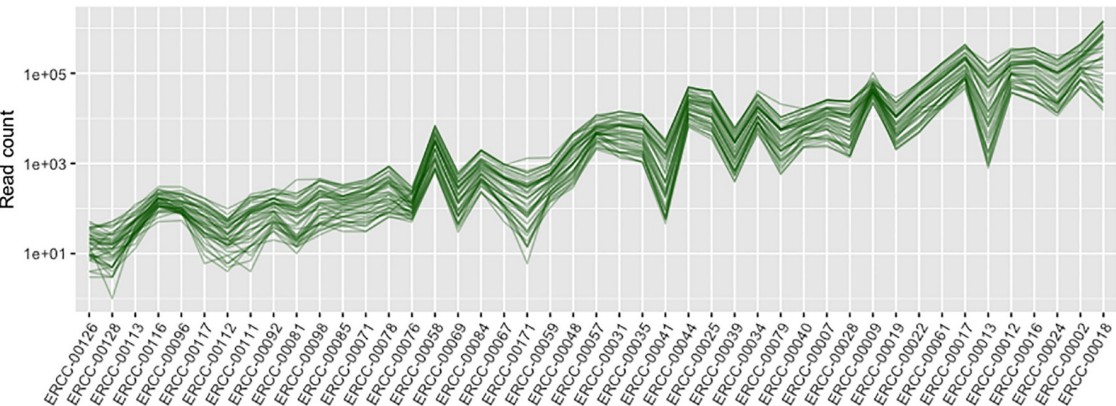

**Fig 1. Correlation of references in experimental data.** Read counts of ERCC spike RNAs in ENCODE mouse tissue samples are plotted on parallel coordinates. Each polyline represents a sample, spike-in RNAs are sorted by their nominal concentration which is specific for each spike, but constant per spike across all samples. The parallel polylines indicate a positive correlation between these spikes.

minimum size of the target subgraph, or favoring larger size subgraphs, one can reduce the chance of picking up differential ones. Second, all reference features are correlated with one another, resulting in a close-to-rank-1 read count matrix. By favoring the subset having smaller rank-1-residuals, one can choose a better set of references. *Rank-1-residuals* of the reference features $R$, more precisely of their read count matrix $C_R$, is the normalized sum of the singular values except the first one.

$$C_R \;\; = \mathbf{U\Sigma V}^T \quad \text{(singular value decomposition of } C_R)$$

$$Rank\text{-}1\text{-}residuals(C_R) \;\; \equiv \frac{1}{\|\sigma\|_2} \sum_{i=2}^{d} \sigma_i^2 \;, \quad [\sigma_1, \sigma_2, \cdots, \sigma_d] \text{ are the singular values of } C_R$$

Altogether, these observations imply that by finding all `maximal_cliques()` and select the `best()` one, i.e. the lowest rank-1-residuals with a large enough size, one can identify the set of references (Algorithm 1) to be used in normalizing the read counts (Algorithm 2).

**Algorithm 1** Graph-based reference identification

```
function IDENTIFY_REFERENCES(C)
  for i from 1 to (n − 1) do
    for j from (i + 1) to n do
      if cor(i, j) ≥ t then E_ij ← 1
    Build graph: G = (V,E)
    candidates = maximal_cliques(G)
  return best(candidates)
```

**Algorithm 2** Graph-based normalization

```
function GBNORM(C)
    Stage 1. Identify references: R ← identify_references(C)
    Stage 2. Normalize C against R:
  for every sample i do
    calculate scale factor s_i ← ∑_{j∈R} C[i, j]
    A[i,j] ← C[i,j]/s_i
  return A
```

**Practical considerations.**  The idea outlined in Algorithm 1 is in fact not efficient enough for realistic data. The first major cost is incurred by the construction of graph $G(V, E)$ which requires $O(|V|^2)$ both in time and space for calculating and storing the correlation matrix, with

$|V|$ being the number of vertices. A typical transcriptome with 70000 features takes up 18GB, far exceeding the average computer memory of 8GB at the time of this writing. In practice, the number of vertices can be significantly reduced by retaining only features that have non-zero (or high enough to be considered reliably detected) read counts across all samples. The cumulative distribution of minimum read count across samples of the real data set revealed that 85% of the vertices can be eliminated with a non-zero filter (S1 Supporting Methods). The vertices can be pruned even further with a low-expression filter, for example, to retain only transcripts that are persistently expressed above the lower quartile. Since RNA-seq measures are most reliable in the moderate expression levels, such trimming practice, in addition to reducing complexity of the graph problem, will also improve reliability of the input and have been a common practice in earlier normalization strategies [2, 3]. The second bottleneck happens at the clique problem. The enumeration of all maximal cliques takes exponential time, for which the most efficient algorithm available runs in time $O(d|V|3^{d/3})$, that is, exponential in graph degeneracy $d$ which measures a graph sparsity, making it efficient only on sparse graphs where $d$ is small enough [18]. Since a feature cannot be both reference and differential at the same time, it is sufficient to find non-overlapping subgraph. This allows us to avoid the prohibitive cost of enumerating all maximal cliques replacing this problem with finding densely connected subgraphs, i.e. graph communities (Algorithm 3). Because we are only concerned with one outstanding community in the graph, an accurate and complete graph partitioning may not be necessary. Furthermore, as a subset of references is sufficient for good normalization (Theorem 2), it is tolerable to miss a few members in the target community. For those reasons, many good graph partitioning methods can be used in this step. Among the available methods, affinity propagation [19] conveniently takes a similarity matrix as an input, thus can be run on correlation matrix without transformation.

For a proof-of-concept implementation, we used the R package `apcluster` [20] for affinity propagation algorithm. Parametric choices were elaborated further in the S1 Supporting Methods.

**Algorithm 3** Graph-based reference identification, with practical considerations

```
function IDENTIFY_REFERENCES(C)
    remove features with zero/low counts, m features remain
  for i from 1 to (m − 1) do
    for j from (i + 1) to m do
      if cor(i, j) ≥ t) then Eᵢⱼ ← t
    Build graph with weighted edges: G = (V,E)
    candidates = community(G)
    remove candidates with minimum cor < t
    return arg min rank_1_residuals(b)
         b∈candidates
```

## Methods

Normalization performance is measured against a collection of mini validation data sets and a full-size data set, where a ground truth can be computed using ERCC spike-ins. The effect of normalization in the presence of batches was observed on a batch-confounded data set.

### Performance measure

Performance of normalization procedure was measured in two terms, by *precision* in detecting the references, and by *condition-number-based deviation* (*cdev*) from the ground-truth normalization. Both measures are only computable when a ground-truth is known. In the case of precision, constant-concentration spike-in RNAs are the ground-truth references. In the case

of *cdev*, the expression matrix resulted from normalizing against these spike-ins serves as the ground-truth normalization.

**Condition-number-based deviation (*cdev*).** Let $A$ be the true abundance matrix. Since $A$ cannot be measured directly, in the setting of performance measure, a ground-truth normalization serves as a surrogate for $A$, i.e, $A \equiv normalize(C, R)$, with $R$ being the set of all true references.

By definition, an ideal normalization is one that can be transformed into $A$ by multiplying with a constant factor, i.e., $A^*$ is ideal normalization if $A^* \times constI = A$, with $I$ being the identity matrix.

Let $A_X \equiv normalize(C, X)$ be result of normalizing $C$ against the set $X$, $B_X$ the matrix that transforms $A_X$ to the ground-truth normalization $A$.

$$
\begin{aligned}
A_X \cdot B_X &= A \\
A_X^T A_X B_X &= A_X^T A \\
B_X &= \left(A_X^T A_X\right)^{-1} \cdot \left(A_X^T A\right)
\end{aligned}
$$

Since any constant multiplication of the ground-truth $A$ is a valid normalization (see Definitions and Notations), $A_X$ is considered valid if $B_X = const \times I$. Hence the quality of the reference set $X$ can be measured by how much $B$ deviates from the identity form, quantified by the condition number of $B$.

$$
\kappa(B_X) = \|B_X\| \cdot \|B_X^{-1}\|
$$

In another word, $X$ is a better reference set and $A_X$ is closer to the ground truth if the condition-number-based deviation of $A_X$ from $A$, denoted $cdev(A_X, A) \equiv \kappa(B_X)$ is closer to 1. For brevity, *cdev* in the following text is implicitly measured against the ground truth normalization $A$.

## Data sets

**Validation data sets.** Transcriptomic profiling data were downloaded from the ENCODE project [21], including 71 samples covering various mouse tissues (Table 1). All of the raw output have been pre-processed according to the ENCODE Uniform Processing Pipeline, i.e. aligned to the mouse genome *mm10*, and quantified by RSEM [22] to result in expected read counts, using the gene/transcript annotation GENCODE *M4*. These results were then consolidated into a read count matrix of 71 samples × 69691 features. This matrix is the *Full* data set. Among these samples, only a subset of 41 were spiked with ERCC synthetic RNAs (at concentrations established in the NIST Pool 14), composing the *Full Spiked* set. Together, these two sets help to evaluate the performance of reference identification in a realistic situation. The *Full* set is typical for what is usually subjected to RNA-seq normalization algorithms: covering the complete transcriptome with a number of internal references. Since this set includes some samples that were not spiked with external RNAs, the spike RNAs are differential, thus should not be selected by reference identification procedure. As internal references are constant across all samples in the *Full* set, they are also constant in the *Full Spiked* set, thus remain the references in this subset. The *Full Spiked* set, as its name implies, has additional references thanks to the spiked RNAs. As stated by Proposition 2, both reference sets should result in equally valid normalization. Since external references are known, normalization by them is chosen to be the ground truth. A reference set is better if it can normalize the read counts closer to the ground truth.

**Table 1. Number of samples per conditions in the *Full* and *Validation* data sets.**

| Condition (Biosample type) | *Full* set | *Full Spiked* / *Reduced* / Mini Validation sets [a] |
|---|---|---|
| adrenal gland | 2 | 0 |
| brain | 2 | 2 |
| central nervous system | 6 | 4 |
| cerebellum | 2 | 2 |
| colon | 2 | 2 |
| cortical plate | 2 | 0 |
| erythroblast | 2 | 0 |
| frontal cortex | 2 | 2 |
| G1E | 2 | 0 |
| G1E-ER4 | 8 | 0 |
| gonadal fat pad | 2 | 2 |
| heart | 2 | 0 |
| kidney | 2 | 2 |
| large intestine | 2 | 2 |
| limb | 2 | 1 |
| liver | 8 | 4 |
| lung | 2 | 2 |
| megakaryocyte-erythroid progenitor cell | 2 | 0 |
| ovary | 3 | 2 |
| placenta | 2 | 2 |
| small intestine | 4 | 2 |
| spleen | 2 | 2 |
| stomach | 2 | 2 |
| testis | 2 | 2 |
| thymus | 2 | 2 |
| urinary bladder | 2 | 2 |
| **Total** | **71** | **41** |

[a] These sets have the same number of samples

A subset of 41 samples that were spiked with ERCC synthetic RNAs (NIST Pool 14 concentrations) and 1433 genes plus spike RNAs were used to construct the validation data set as illustrated in Fig 2B. In these data, the ERCC spike RNAs serve as reference features, while the differential features are emulated by genes known to participate in signal transduction pathways (REACTOME accession R-MMU-162582 [23]). This choice of differential genes aims to ensure that the validation data set (1) covers a wide range of expression levels (gene products in a signaling cascade are expressed at different levels), (2) includes biologically meaningful correlation, i.e. regulated co-expression, besides artifact correlation of the reference genes and (3) mimics the variability across tissue types (signaling pathways are generally different across biological conditions) (see S1 Supporting Methods). From this pool of differential features, mini validation sets are generated by picking random combinations of signal transduction pathways (Fig 2B).

**Batch-confounded data set.** To illustrate the role of normalization in batch-confounded data, we used a set of 26 samples including 13 pairs of comparable human—mouse tissues. This data was originally collected as part of the ENCODE project [24], and later re-analyzed by Gilad and Mizrahi-Man [25]. Re-analysis showed that the data was confounded by batches, resulting in samples being clustered by species. RNA-seq read counts of these samples and a

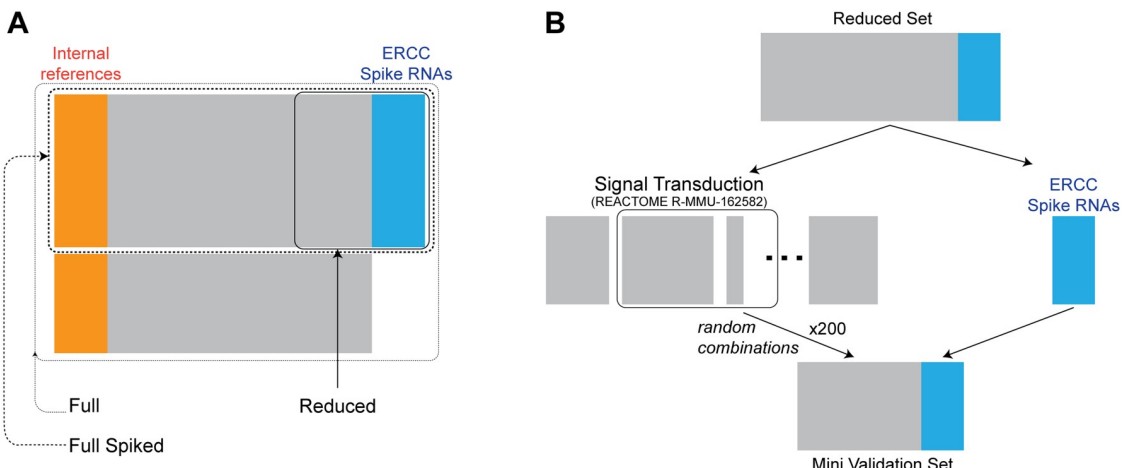

**Fig 2. Diagram of validation data set construction.** (**A**) The largest—*Full* dataset—includes 71 samples (rows) and 69691 features (columns). A subset of these samples were spiked with ERCC spike-ins, making up the *Full Spiked* data set (41 samples × 69691 features). Among these features (i.e. genes), only those those involving the signal transduction pathways were selected to create the *Reduced* set from which mini validation sets were sampled from. (**B**) Method of generating mini validation sets from experimental mRNA-seq data. The *Reduced* set includes 1433 genes from signal transduction pathways, plus all ERCC spike-ins. From this pool, 200 mini validation sets were created by random selection of 08 out of 16 pathways.

table of human-mouse orthologous gene pairs were downloaded at doi:10.5281/zenodo.17606 [26]. From these pre-processed counts and gene list, an expression matrix of 26 samples × 14646 genes was compiled.

Batch effects were corrected using ComBat [27] (implemented in the R package sva [28]) to adjust for five sequencing batches identified by Gilad and Mizrahi-Man [25, 26]. ComBat was run either on the raw counts or that normalized by various methods, including gbnorm, resulting in different corrected versions of the expression matrix. Genes with levels in the lowest 30% quantile were then removed, and samples were clustered based on distances calculated from the remaining genes. Various clustering methods (hierarchical, spectral, k-means) on different distance metrics were attempted. The result of hierarchical clustering on Euclidean distance with UPGMA agglomeration scheme is reported in the main text, while the rest of this analysis and source code are available in S2 R Notebook.

## Results

### Normalization of validation data

The validation data sets include 200 mini expression matrices created as described in Fig 2. The performance of *gbnorm* is compared with random subset of references and with some existing normalization methods including UQ [2], TMM [3], DESeq [4] and PoissonSeq [5].

Two parameters of the graph construction step were explored: correlation method (Pearson and Spearman) and transformation on the read count (identity, $\log_2$ and $\log_{10}$). References were identified with high precision ($>0.8$) on the mini validation sets, regardless of the graph-construction parameters, resulting in significantly better normalization results compared to random selection of the reference sets. Log-transformation and Pearson Correlation Coefficient (PCC) tend to result in slightly better reference sets, with higher precision ($>0.9$) and consistently lower *cdev* (Fig 3).

In comparison with earlier normalization methods, *gbnorm* on validation sets always resulted in normalized counts closer to the ground-truth obtained by normalizing against the external spikes (Fig 4).

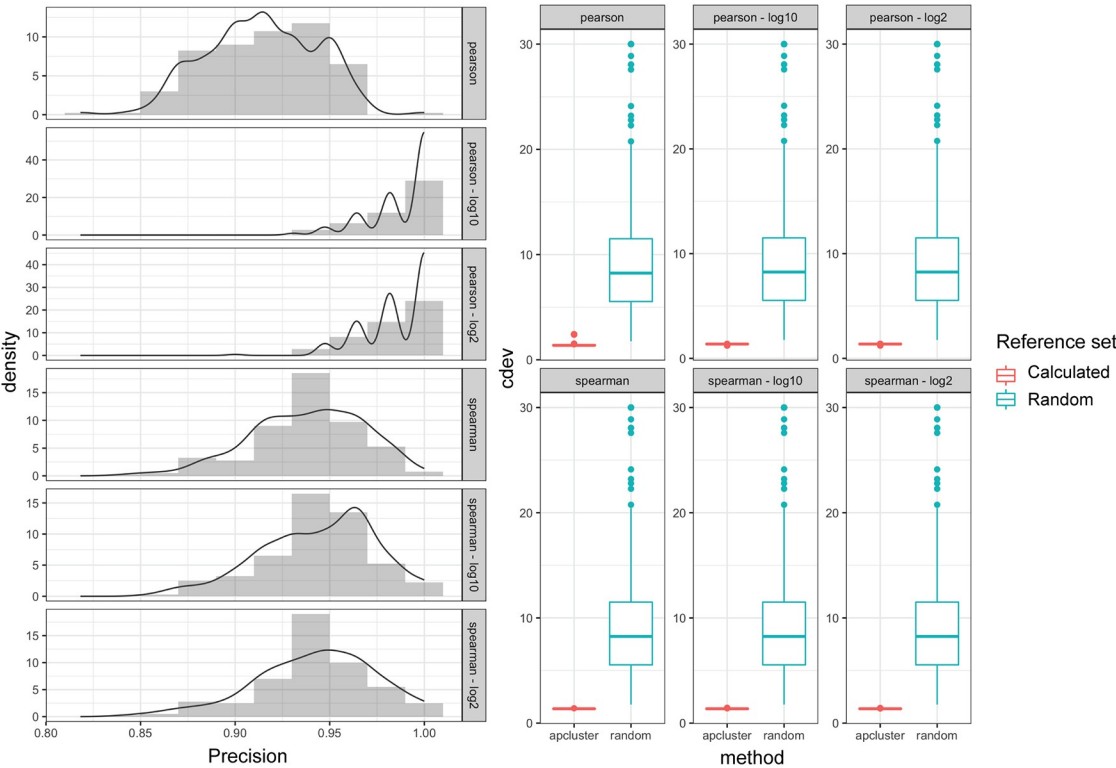

**Fig 3. Performance of graph-based normalization on the validation data measured by precision in reference identification (left) and by deviation from the ground-truth normalization, $cdev(A_X, A^*)$ (right).** $cdev(A_{random}, A^*)$ resulted from normalization by random set of references are plotted for comparison. Three types of transformation (identity, $\log_2$, $\log_{10}$) and two correlation measures (Spearman, Pearson) were tested, resulted in six different configurations of input to the reference identification procedure.

## Normalization of real data

Graph-based normalization may have difficulty detecting references in a real data set where there are only internal references (genes that are naturally expressed at the same level across conditions) but no external ones (exotic RNAs added at constant concentration to all samples). To test its applicability in such realistic situation, we performed reference identification on the *Full* set, and measured the quality of its output on the *Full Spiked* where it is possible to calculate a ground-truth normalization.

*gbnorm* was run in two stages, (1) reference identification and (2) scaling. Such split-up of the algorithm was to test its ability to detect references in practice when there is no artificially constant RNAs, while retaining our ability to measure normalization quality against a ground-truth dictated by external spikes. The first step was run on both the *Full* set and the *Full Spiked*, with one of the best graph-construction parameters as determined on the validation data sets, i.e. edges were formed by Pearson correlation on the log-transformation of read counts. The later used output from the first to normalize the *Full Spiked* set. Other normalization methods were run in one single step to normalize the *Full Spiked* set. As shown in Table 2, *gbnorm* was able to identify internal references where there were no spikes, resulting in a better (smaller *cdev*) normalization compared to other methods. While earlier methods can be done within seconds, graph-based normalization is inherently expensive, taking from a few to 30 minutes depending on the input on the current data set. This running time is still reasonably accessible for one-time operations. On the *Full Spiked* set, *gbnorm* based on affinity propagation clustering had trouble converging to a good solution, resulting in a long running time and worse

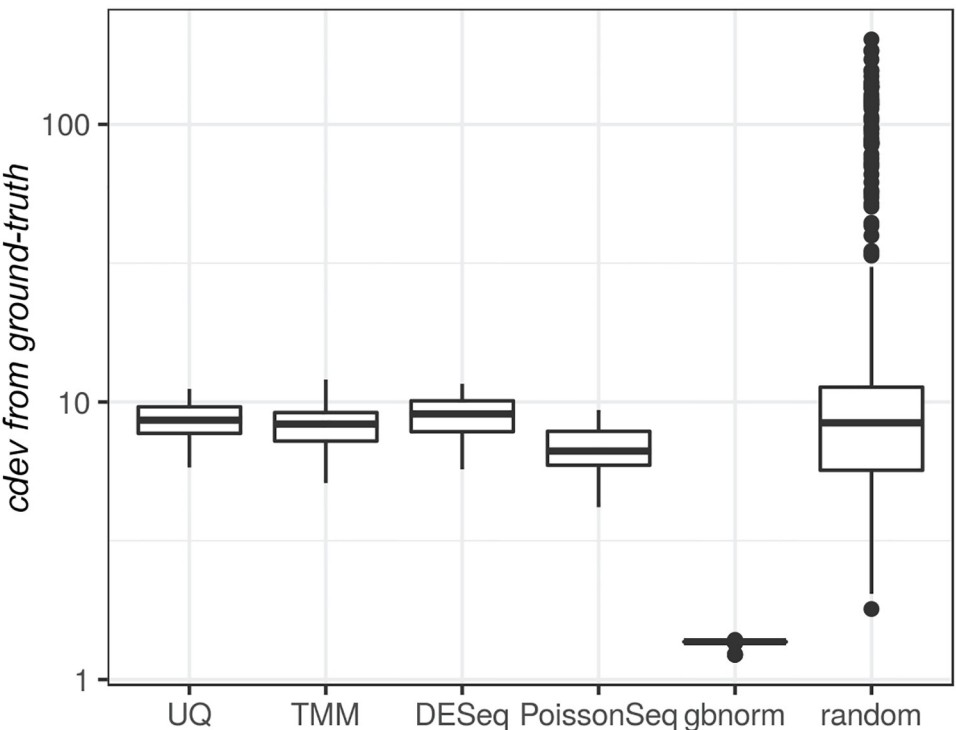

**Fig 4. Comparison of graph-based normalization and existing global-scaling methods.** Distribution of deviation from the ground-truth, measured by *cdev*, when normalizing the validation sets by UQ, TMM, DESeq, PoissonSeq, gbnorm and by random sets of features. Ground-truth is obtained by normalizing against ERCC spike-ins with at least 100 counts).

normalization. It remains to be studied whether other clustering methods can improve on this data set, and how sensitive they are with respect to different properties of the input data such as the correlation distribution and the community structure.

## Normalization on batch-confounded data set

Given the widespread presence of batch effects [29, 30] in high-throughput data, RNA-seq analytic workflows typically include a batch correction step following normalization [31]. To test the ability of *gbnorm* to facilitate batch effect removal, we observed the clustering pattern of samples in a human—mouse tissue data set. This data set was originally published by Lin

**Table 2. Comparison of *gbnorm* and existing normalization methods.**

| Method | Reference Identification set | Scaled set | Running time (seconds) | cdev from ground-truth |
|---|---|---|---|---|
| raw | NA | Full Spiked | 0.006 | 17.416850 |
| UQ | NA | Full Spiked | 0.519 | 7.189026 |
| TMM | NA | Full Spiked | 1.085 | 7.952690 |
| DESeq | NA | Full Spiked | 0.454 | 8.439418 |
| PoissonSeq | NA | Full Spiked | 0.174 | 8.007819 |
| gbnorm | Full | Full Spiked | 121.565 | 6.058230 |
| gbnorm | Full Spiked | Full Spiked | 1329.980[†] | 8.943961 |

[†] Long running-time due to affinity propagation not converging

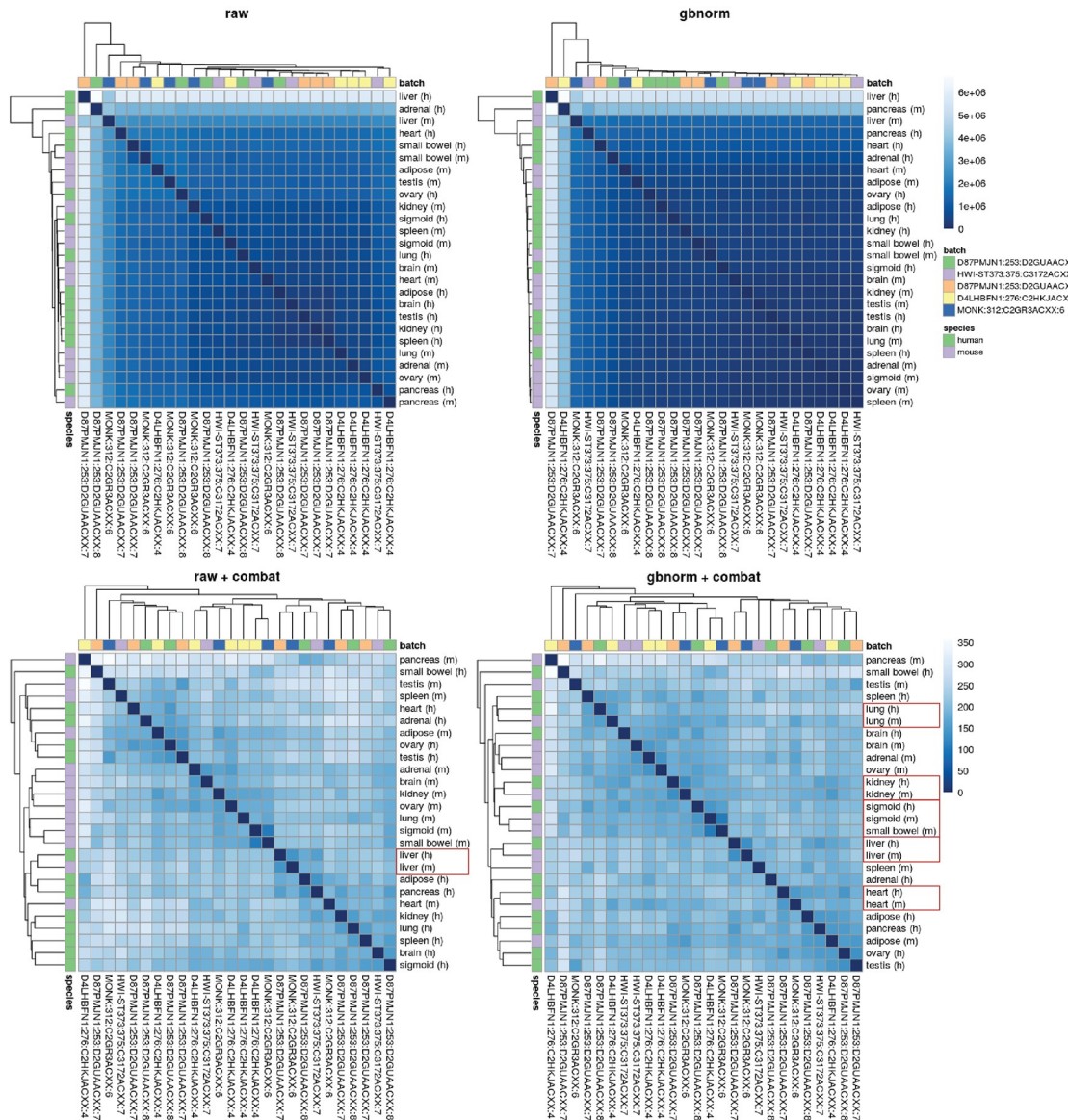

**Fig 5. Batch effect removal performed on raw vs normalized counts resulted in different clustering pattern of the samples.** No processing (**raw**) or normalization alone (**gbnorm**) did not create clear groups of samples, batch effect correction alone (**raw + combat**) resulted in samples clustered mostly by species, while the combination of normalization and batch effect correction (**gbnorm + combat**) resulted in samples clustered mostly by tissues. Samples were clustered by hierarchical clustering, with UPGMA agglomeration method, on Euclidean distance calculated from genes above the 30% quantile.

et al. [24] and later demonstrated to be confounded by sequencing batches such that uncorrected data resulted in samples being clustered mostly by species [25]. Here we performed clustering on four different versions of the expression matrix: uncorrected raw counts, uncorrected normalized counts, batch-corrected raw counts and batch-corrected normalized counts. Without batch effect removal, expression profiles of the tissue samples were barely distinguishable, with some outliers standing out from the rest of samples (Fig 5). In contrasts, batch effect removal using ComBat [27] resulted in distinct groups of samples, with grouping pattern varies depending on how the inputs were processed. Correcting for batch effects on raw counts resulted in the clustering of samples mostly by species, while on

normalized counts resulted in clustering mostly by tissues (Fig 5). This shift of clustering pattern was observed for all normalization methods (S2 R Notebook), suggesting that normalization, in general, is helpful for batch effect removal, and that most scaling methods examined here (UQ, TMM, DESeq, PoissonSeq, and gbnorm) are equivalent in this role. While different normalization methods resulted in only slightly different clusters, depending on the choice of clustering algorithms and distance metric, batch-corrected raw counts consistently failed to group samples by tissues.

## Discussion

Using a simplistic treatment of RNA-seq read counts, we showed that artificial correlations are theoretically expected, and experimentally observed in RNA-seq data. This observation leads to a new way to identify reference features, and consequently, to normalize RNA-seq read counts. Unlike existing methods, this algorithm helps identify a set of references, based on their correlation in read counts, thus eliminating the need for prior knowledge about stably expressed genes and assumptions about similar expression profiles between experimental conditions. It's worth noticing that this method requires a large number of samples and a quality feature count method for reliable correlation measure.

Upon the finding on artifactual correlations among reference features, the challenge of performance measure is renewed. Quality of RNA-seq normalization has been traditionally judged on the accuracy of DEG detection which requires simulated data sets where DE and non-DE genes can be specified. As conventional simulation procedures do not capture the correlations in the real data, we constructed validation data sets from real measurements, eliminating the need to simulate sequencing reads while preserving the intrinsic correlation required for graph-based normalization. Although this construction aims to measure the performance of graph-based normalization (hereafter denoted as *gbnorm*), it was crafted to introduce more adversities rather than advantage for the method. This construction also allows us to use a new performance measure based on comparison against the ground-truth instead of relying on the result of DEG detection, effectively uncoupling the normalization step from its downstream analyses. Even though such uncoupling prevents us from positing further on the performance of *gbnorm* in these analyses, we believe it is a necessary restriction, for those assessments requires careful selection of data sets and performance measures specific to the biological insights being pursued. More sophisticated simulation protocols to re-produce the real data in target properties while retaining correlation structure may be necessary to study such downstream effects of normalization.

To avoid confusion, a discussion on the broad meaning of *normalization* in the gene expression literature has been delayed until now. Although *normalization* in this article has been formally defined, and so far exclusively meant, to be a *global scaling* operation in which a single scaling factor is used per sample, the term may also indicate other distinctive transformations of read count matrix. In particular, *factor analysis* such as RUV [32] allows for both sample- and gene-specific scaling factors, and *quantile normalization* [33, 34] enforces identical distribution of read counts between all conditions, via complete replacement of numerical counts by rank-equivalent counts. These processes generate highly different normalized results which would be meaningless to compare directly. It is most likely that specific requirements of downstream analyses will dictate which approach is appropriate, and in case they all are, in which terms they can be compared. That said, these approaches are not always mutually exclusive. For instance, the RUVg variant of RUV method relies on a set of references, either known or determined empirically, to factor out unwanted variations. The empirical determination of these references relies on other processes, such as DE tests with built-in global scaling

normalization, or graph-based reference identification. This way, RUV and global scaling methods are complementary steps in a normalization protocol.

Another important aspect of gene expression profiling analysis is batch effect removal. Batch effects can arise from various technical discrepancies in the whole experimental process, ranging from instruments, reagents, protocols to handlers and time, producing non-biological sources of variations [29, 30]. The use of ERCC spike-ins to establish the ground truth may lead to an expectation that this ground truth is free from all batch effects. This expectation would be reasonable if spike-ins are used to derive a standard curve from which absolute RNA concentrations can be inferred, a use originally intended for these synthetic sequences [11]. Other than building standard curve, spike-ins can also be used in global scaling and factor analysis, to derive sample-specific and sample-gene-specific scaling factors, respectively [35]. Since batch effects can be complicated, e.g. affecting different genes in different ways [29], it is apparent that sample-specific scaling factors will not be able to eliminate all the unwanted variations. That realization does not negate the need of global scaling normalization. As indicated by our analysis on a batch-confounded data set, such normalization prior to batch effect correction can critically improve the outcome. Still, the complete remedy of batch effects is a complicated matter which calls for scrupulous planning as early as experimental design, as well as deliberations on batch effect removal algorithms [30].

A peripheral, yet important implication of this work is the fact that artifactual correlations do arise from RNA sequencing process. Because correlations have been routinely used to measure co-expression and to derive biological meanings [36, 37], this finding suggests that researchers exercise more caution when interpreting correlations on RNA-seq read counts. As a result of Observation 1, correlation does not always indicate the existence of biological regulation.

Graph-based normalization promises to deliver a better result, evidently at the cost of higher computational demand. In many conventional use cases such as DE detection in comparable conditions, it may still be preferable to use conventional methods such as TMM [3] and DESeq [4, 38], which performed well in previous benchmark studies [14]. As the need arises for analyzing more expression profiles of higher heterogeneity, methods without assumption about condition similarity such as *gbnorm* should be considered. The ability to work on abundant and heterogeneous profiles make it more versatile, applicable to a wide range of settings such as mRNA-seq from different organs, tissue types, developmental stages, cell types, or even 16S-RNA-seq from diverse microbiomes. Analytic methods that specifically required normalized expression levels as their input [39, 40] will benefit from cross-sample, application-independent normalization methods such as *gbnorm*. Although presented here as a normalization method, at the core of *gbnorm* is a reference identification procedure. These references can be applied in different ways. In many count-based DE tests, it is critical that the input count matrix remain integer, and normalization factors introduced as offsets. Users of these workflows should be highly aware of how scaling factors are defined and used in each case [41, 42].

*Gbnorm* relies on a community detection algorithm which has plenty of options from previous research, provided that the chosen algorithm does not require the number of communities as an input. Unfortunately, this wealth of choices implies that the performance can vary significantly depending on specific community detection approach and its corresponding parameters, as is the case with many graph-based problems. Our current choice, the affinity propagation algorithm, conveniently takes similarity matrix as an input, avoiding the need to transform similarity into distance measure. Whether other algorithms provide better performance remained to be explored, although they may introduce a varying set of parameters. Beyond algorithmic parameters, it is also important to understand how the method perform in

various practical settings, specifically gene-level *vs* transcript-level read counts, the degree of heterogeneity among conditions, and the proportion of differential features. Such knowledge will guide the development of a more versatile and faster normalization tool.

In summary, as a proof-of-concept, it is an encouraging result that we may be able to break the circular dependency problem in RNA-seq data. Further studies in the directions outlined above are needed toward more practical implementations.

## Supporting information

**S1 Supporting Methods. Characterization of the data sets and parametric details.** (PDF)

**S1 R Notebook. Interpreting condition number as the variance of random noise.** (HTML)

**S2 R Notebook. Clustering of human and mouse tissue samples.** (HTML)

## Acknowledgments

The authors would like to thank Dr Jay Gertz for insightful discussions, Nghia Truong for the math discussions, and Dr Jeff Phillips for helping to improve the early version of this work. We thank the anonymous reviewers for their valuable comments and suggestions.

## Author Contributions

**Conceptualization:** Diem-Trang Tran.

**Data curation:** Diem-Trang Tran.

**Formal analysis:** Diem-Trang Tran.

**Funding acquisition:** Balagurunathan Kuberan, Matthew Might.

**Investigation:** Diem-Trang Tran.

**Methodology:** Diem-Trang Tran, Aditya Bhaskara.

**Software:** Diem-Trang Tran.

**Supervision:** Matthew Might.

**Validation:** Diem-Trang Tran, Aditya Bhaskara, Matthew Might.

**Writing – original draft:** Diem-Trang Tran.

**Writing – review & editing:** Diem-Trang Tran, Aditya Bhaskara.

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
