## [Decision Letter · Decision Letter 0]

6 Aug 2019

PONE-D-19-17038

A graph-based algorithm for RNA-seq data normalization

PLOS ONE

Dear Ms Tran,

Thank you for submitting your manuscript to PLOS ONE. After careful consideration, we feel that it has merit but does not fully meet PLOS ONE’s publication criteria as it currently stands. Therefore, we invite you to submit a revised version of the manuscript that addresses the points raised during the review process.

Both reviewers felt that an improved discussion on how the method responds to batch effect and more analysis of of how the method works as compared to other similar techniques. Improvements to these issues would greatly help your paper. Please respond to the reviewers observations and take care to address the following issues:

Better analysis of how the method deals with batch effect, as mentioned in the comments from reviewers 1&2. Include comparisons to other comparable methods, such as RUVSeq and packages that can determine batch effects such as sva and ComBatImprove paper layout, as requested by reviewer #2

We would appreciate receiving your revised manuscript by Sep 20 2019 11:59PM. To enhance the reproducibility of your results, we recommend that if applicable you deposit your laboratory protocols in protocols.io, where a protocol can be assigned its own identifier (DOI) such that it can be cited independently in the future. For instructions see: http://journals.plos.org/plosone/s/submission-guidelines#loc-laboratory-protocols

We look forward to receiving your revised manuscript.

Kind regards,

Kyle Ellrott

Academic Editor

PLOS ONE

Journal Requirements:

1. We note that you have stated that you will provide repository information for your data at acceptance. Should your manuscript be accepted for publication, we will hold it until you provide the relevant accession numbers or DOIs necessary to access your data. If you wish to make changes to your Data Availability statement, please describe these changes in your cover letter and we will update your Data Availability statement to reflect the information you provide.

Additional Editor Comments (if provided):

Thank you for your submission. The reviewers have raised several questions about some of the analysis and formatting of the paper. Both reviewers raised questions about how the method would deal with batch effect and other technical biases. Please address these issues before resubmitting.

Reviewers' comments:

Reviewer's Responses to Questions

**Comments to the Author**

1. Is the manuscript technically sound, and do the data support the conclusions?

Reviewer #1: Partly

Reviewer #2: Partly

2. Has the statistical analysis been performed appropriately and rigorously? 

Reviewer #1: Yes

Reviewer #2: No

3. Have the authors made all data underlying the findings in their manuscript fully available?

Reviewer #1: Yes

Reviewer #2: Yes

4. Is the manuscript presented in an intelligible fashion and written in standard English?

Reviewer #1: Yes

Reviewer #2: Yes

5. Review Comments to the Author

Reviewer #1: I believe that this is a novel and interesting approach to gene expression normalization. While I don't think there is anything theoretically incorrect with your implementation, I believe there should be one more comparative method and at least one additional metric (independent of spike-in values) with which you score your model to ensure it works in practice.

Here are my two main concerns with the manuscript:

1) Dependence on spike-in values for evaluation. Your method does not adequately show that your algorithm addresses the problem that normalization attempts to answer - to remove unwanted variation - i.e. variation that is irrelevant to the experiment performed. In this paper you use spike-ins as the gold standard, however it has been reported many times spike-ins have a high variability within replicates (Qing. et. al 2013 "mRNA enrichment protocols determine the quantification characteristics of external RNA spike-in controls in RNA-Seq studies."). I believe that your method may still be useful, but the metrics you use are only related to reconstructing spike-in expression, or identifying the spike-ins as controls. This could potentially be very misleading, since reconstructing the spike-in values could potentially be biased. It would be much more convincing if you could include another metric that is independent of the spike-ins/control genes, but shows that the downstream analyses are improved using your method vs. other methods. One such example would be to show that your method removes technical variation, such as batch effects. This has been nicely done on ENCODE expression data, which you already have experience with, by showing that the gene expression of each tissue clusters with one another instead of by species (mouse vs. human). This has been discussed in (Gilad et. al. 2015 "A reanalysis of mouse ENCODE comparative gene expression data"). If your method was to outperform their normalization, in terms of clustering tissues more closely together instead of species, this would be more convincing. Two metrics commonly used to judge clusters are the adjusted rand index or silhouette score.

2) There currently exists a method very similar to your proposed method called RUVSeq (Risso et. al. 2015 "Normalization of RNA-seq data using factor analysis of control genes or samples"), which also uses control genes (spike-ins or computationally identified) to normalize the data. However it differs in the implementation: Risso et. al. uses a linear modeling approach. Since this method is very similar and it is used in the field (531 citations), I believe it is the most relevant method to compare against. It may also be useful to evaluate your method in a similar way that RUVSeq does.

At the minimum I believe you should at least temper your findings and put them in context with other, techniques out there for normalization that have been used with great success on large datasets, especially in quantitative trait loci studies (see GTEx, TCGA, ICGC, and ENCODE).

Minor comments:

- You have a section called "Preliminary results", why are these preliminary?

- Can you share the code for gbnorm? The details are available, but having the code will make it much more widely used, perhaps putting it on bioconductor would be useful or maybe a R notebook.

- A very good use case for this data would be in cancer studies, where the assumption that a majority of the genes are not differential can be violated. It may be useful to do one differential expression analysis on TCGA count data on a large dataset that has a known artifact (sampled by two labs, or a later release of the TCGA data)

- In Table 2 the result doesn't seem very convincing given the drastic increase in runtime. Maybe this is because I don't really have an intuitive sense of the magnitude of cdev. If you could include an illustration of how a difference of ~1 cdev is significant for downstream analyses this can also improve the impact of the paper.

Reviewer #2: This paper introduced a graph-based algorithm for RNA-Seq data normalization without exclusion of differentially expressed genes, however, the manuscript was poorly written and the algorithm was not clearly explained. I also have the following major comments.

1. I disagree with the authors' statement that 'inclusion of differential feature in normalizing factors leas to invalid normalization'. Normalization does not remove biological variation or phenotype effects if these factors or conditions are separated from technical artifacts by a valid statistical model. For example, Bioconductor package sva detects latent technical batch effect on RNA-seq counts in sva() and removes the noise via ComBat().

2. The author suggested to use a subset of references for normalization. What does a subset reference mean in this paper? Is it mRNA only, non-coding RNA only, spike-in RNA only? The distribution of mRNA, non-coding RNA transcripts abundance are quite different, hence, differential expression patterns for mRNA and non-coding RNA genes are different. See a recent paper 'lncDIFF: a novel quasi-likelihood method for differential expression analysis of non-coding RNA (BMC Genomics)'.

3. I suggest the authors to reformat the manuscript as introduction, methods, results, discussion, conclusion. The current format is difficult to read. The statements in Theorem, Observations should be merged with other paragraphs. I don't see the necessity to present the 'proofs' in the main text. Besides, these statements are not exactly mathematical theorems and the equations (1)-(6) should not a labeled as 'proof'.

6. PLOS authors have the option to publish the peer review history of their article (what does this mean?). If published, this will include your full peer review and any attached files.

Reviewer #1: No

Reviewer #2: No

---

## [Author Response · Author response to Decision Letter 0]

7 Oct 2019

A document titled "Response to Reviewers" with rich format is uploaded as part of this submission.

---

## [Decision Letter · Decision Letter 1]

5 Nov 2019

PONE-D-19-17038R1

A graph-based algorithm for RNA-seq data normalization

PLOS ONE

Dear Ms Tran,

Thank you for submitting your manuscript to PLOS ONE. After careful consideration, we feel that it has merit but does not fully meet PLOS ONE’s publication criteria as it currently stands. Therefore, we invite you to submit a revised version of the manuscript that addresses the points raised during the review process.

Please address the specific issues raised by reviewer #2 regarding the underlying assumptions in equations 1 & 2 as well as the notes on performance metrics.

We would appreciate receiving your revised manuscript by Dec 20 2019 11:59PM. To enhance the reproducibility of your results, we recommend that if applicable you deposit your laboratory protocols in protocols.io, where a protocol can be assigned its own identifier (DOI) such that it can be cited independently in the future. For instructions see: http://journals.plos.org/plosone/s/submission-guidelines#loc-laboratory-protocols

We look forward to receiving your revised manuscript.

Kind regards,

Kyle Ellrott

Academic Editor

PLOS ONE

Additional Editor Comments (if provided):

While the reviewer felt that this new revision covered a majority of the issues found in the original draft, there are still two technical points that need to be clarified.

1) Reviewer #2 asserts that the underlying assumption in equations 1 and 2 do not make them equivalent: " Equations (1) and (2) are NOT equivalent, because c_ij /M_i is NOT proportional (or equal) to a^rel_ij. "

2) Reviewer #2 asserts that the assumptions made for normalization and batch effect correction performance metrics were incorrect: "The performance metrics did not measure the removal of batch effect, due to the incorrect assumption a*_ij=constant * a_ij."

Reviewers' comments:

Reviewer's Responses to Questions

**Comments to the Author**

1. If the authors have adequately addressed your comments raised in a previous round of review and you feel that this manuscript is now acceptable for publication, you may indicate that here to bypass the “Comments to the Author” section, enter your conflict of interest statement in the “Confidential to Editor” section, and submit your "Accept" recommendation.

Reviewer #1: All comments have been addressed

Reviewer #2: All comments have been addressed

2. Is the manuscript technically sound, and do the data support the conclusions?

Reviewer #1: Yes

Reviewer #2: Partly

3. Has the statistical analysis been performed appropriately and rigorously? 

Reviewer #1: Yes

Reviewer #2: No

4. Have the authors made all data underlying the findings in their manuscript fully available?

Reviewer #1: Yes

Reviewer #2: Yes

5. Is the manuscript presented in an intelligible fashion and written in standard English?

Reviewer #1: Yes

Reviewer #2: Yes

6. Review Comments to the Author

Reviewer #1: The authors have taken all comments and addressed them thoroughly. The use of the word "normalization" I feel could be addressed earlier in the manuscript, but I think it is clear enough in the discussion. The addition of the notebook to help readers get a feeling for cdev was a very useful addition. Furthermore, the additional experiments also help the reader to get a feeling for how their method would work in a real-world scenario.

Reviewer #2: The authors have revised the manuscript thoroughly and the methods and algorithms are now clearly described. However, there are major errors or concerns in the proof for Theorem 1 and the selection of performance metrics, being a red flag for the validity of this algorithm.

1. Equations (1) and (2) are NOT equivalent, because c_ij /M_i is NOT proportional (or equal) to a^rel_ij. c_ij is the read counts computed by RNA-Seq bioinformatics software, which aligns and maps the reads to genome reference and may result in random discrepancy in aligning and mapping. Hence, c_ij is not a value by multiplying a_ij with a constant not related to i or j. The author did not assume c_ij as a variable randomly deviating from the true abundance a_ij, which was problematic and lack of statistical validity. This also indicates that the statement ' a*_ij=constant * a_ij' may not be valid.

2. The performance metrics did not measure the removal of batch effect, due to the incorrect assumption a*_ij=constant * a_ij. The best normalization performance metric is coefficient of variation (CV) for normalized counts of reference features.

7. PLOS authors have the option to publish the peer review history of their article (what does this mean?). If published, this will include your full peer review and any attached files.

Reviewer #1: No

Reviewer #2: No

---

## [Author Response · Author response to Decision Letter 1]

12 Dec 2019

A rebuttal letter is provided in a separate document.

---

## [Decision Letter · Decision Letter 2]

30 Dec 2019

A graph-based algorithm for RNA-seq data normalization

PONE-D-19-17038R2

Dear Dr. Tran,

We are pleased to inform you that your manuscript has been judged scientifically suitable for publication and will be formally accepted for publication once it complies with all outstanding technical requirements.

With kind regards,

Kyle Ellrott

Academic Editor

PLOS ONE

Additional Editor Comments (optional):

Thank you for your additional revisions. With the additional citations and text the reviewers have been satisfied with the work.

Reviewers' comments:

Reviewer's Responses to Questions

**Comments to the Author**

1. If the authors have adequately addressed your comments raised in a previous round of review and you feel that this manuscript is now acceptable for publication, you may indicate that here to bypass the “Comments to the Author” section, enter your conflict of interest statement in the “Confidential to Editor” section, and submit your "Accept" recommendation.

Reviewer #2: All comments have been addressed

2. Is the manuscript technically sound, and do the data support the conclusions?

Reviewer #2: Yes

3. Has the statistical analysis been performed appropriately and rigorously? 

Reviewer #2: Yes

4. Have the authors made all data underlying the findings in their manuscript fully available?

Reviewer #2: Yes

5. Is the manuscript presented in an intelligible fashion and written in standard English?

Reviewer #2: Yes

6. Review Comments to the Author

Reviewer #2: (No Response)

7. PLOS authors have the option to publish the peer review history of their article (what does this mean?). If published, this will include your full peer review and any attached files.

Reviewer #2: No

---

## [Editor Report · Acceptance letter]

10 Jan 2020

PONE-D-19-17038R2

A graph-based algorithm for RNA-seq data normalization

Dear Dr. Tran:

I am pleased to inform you that your manuscript has been deemed suitable for publication in PLOS ONE. Congratulations! Your manuscript is now with our production department.

With kind regards,

on behalf of

Dr. Kyle Ellrott

Academic Editor

PLOS ONE